# Smoothie: Smoothing Diffusion on Token Embeddings for Text Generation

**Alexander Shabalin** [1 2]   **Viacheslav Meshchaninov** [1 2]   **Dmitry Vetrov** [1]

## Abstract

Diffusion models have achieved state-of-the-art performance in generating images, audio, and video, but their adaptation to text remains challenging due to its discrete nature. Prior approaches either apply Gaussian diffusion in continuous latent spaces, which inherits semantic structure but struggles with token decoding, or operate in categorical simplex space, which respect discreteness but disregard semantic relation between tokens. In this paper, we propose Smoothing Diffusion on Token Embeddings (SMOOTHIE), a novel diffusion method that combines the strengths of both approaches by progressively smoothing token embeddings based on semantic similarity. This technique enables gradual information removal while maintaining a natural decoding process. Experimental results on several sequence-to-sequence and unconditional generation tasks demonstrate that SMOOTHIE outperforms existing diffusion-based models in generation quality. Furthermore, ablation studies show that our proposed diffusion space yields better performance than both the standard embedding space and the categorical simplex. The code is available at https://github.com/ashabalin/smoothie.

## 1. Introduction

Diffusion models attracted a lot of attention in recent years as they show very high generation quality in image (Rombach et al., 2022; Podell et al., 2023), audio (Evans et al., 2024) and video (Blattmann et al., 2023) domains surpassing all previous approaches such as GANs (Goodfellow et al., 2014) and Normalizing Flows (Rezende & Mohamed, 2015). Diffusion models work by introducing a forward

process that gradually degrades an object by injecting Gaussian noise into it, and then learning the reverse process by denoising the object.

Applying diffusion models to text is challenging due to its discrete nature. Nevertheless, several works have explored ways to design suitable diffusion processes. One line of research proposes gradually removing information by replacing tokens with others sampled from a categorical distribution (Austin et al., 2021; He et al., 2023; Lou et al., 2024). Another approach applies Gaussian diffusion to the latent space of token embeddings (Li et al., 2022; Gong et al., 2023a). Additionally, some studies leverage the discreteness of text by performing diffusion directly on the vocabulary probability simplex instead of the embedding space (Han et al., 2023; Karimi Mahabadi et al., 2024).

Each of the described methods offers distinct advantages and limitations, as summarized in Table 1. Gaussian diffusion progressively removes semantic information: under the Euclidean semantic space hypothesis (Hashimoto et al., 2016), the distinguishability of noisy tokens depends on their initial distances in the latent space. The addition of Gaussian noise gradually disrupts these distances, making the semantics of a latent representation increasingly difficult to recover. However, Gaussian diffusion does not account for the discrete nature of text, which complicates the mapping of generated latent vectors back to discrete tokens (Li et al., 2022; Shabalin et al., 2025).

On the other hand, categorical and simplex-based diffusion methods naturally preserve the discreteness of text and eliminate the need for an explicit decoding step. Nevertheless, they disregard semantic relationships between tokens during the noising process, resulting in a more erratic and less meaningful degradation of information.

In this paper, we propose SMOOTHIE, a smoothing diffusion framework that satisfies both properties. We represent each token with a vector based on distances between token embeddings. During the forward process, our diffusion mechanism gradually perturbs these distances, progressively dissolving semantic information. Like simplex diffusion, our method enables natural decoding from latent representations back to tokens. In theory, SMOOTHIE is applicable not only to text, but to any domain where data comes from a categorical distribution with inherent similarity between

---

[1]Constructor University, Bremen, Germany [2]HSE University, Moscow, Russia. Correspondence to: Alexander Shabalin <amshabalin@hse.ru>.

*Proceedings of the $43^{rd}$ International Conference on Machine Learning*, Seoul, South Korea. PMLR 306, 2026. Copyright 2026 by the author(s).

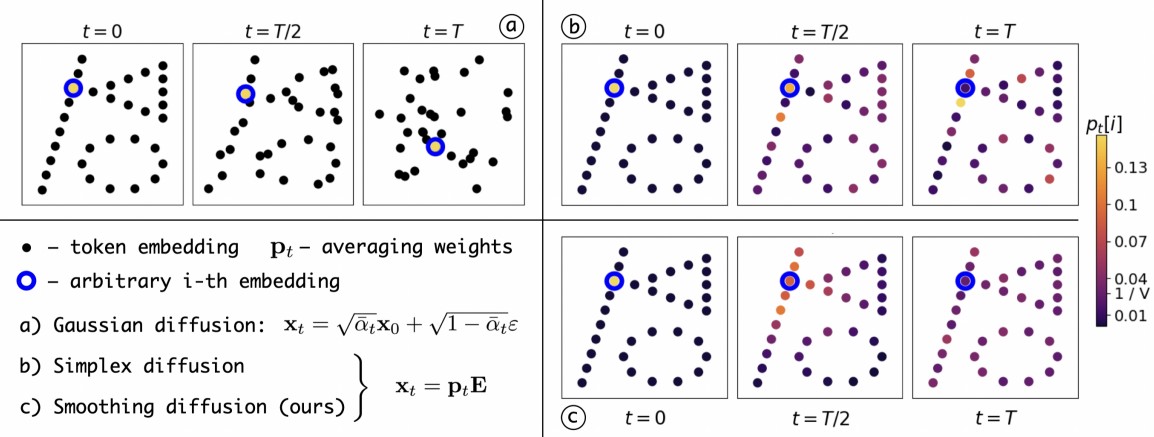

*Figure 1.* An illustration of the diffusion process for Gaussian, simplex, and smoothing diffusion methods. The key distinction between simplex and smoothing diffusion is that the latter incorporates semantic relationships between tokens during the noise addition process.

*Table 1.* Comparison of diffusion methods in terms of accounting for text discreteness and semantics.

|  | Categorical | Gaussian | Simplex | Smoothing (ours) |
|---|:---:|:---:|:---:|:---:|
| Accounting for Discreteness | ✓ | ✗ | ✓ | ✓ |
| Accounting for Semantics | ✗ | ✓ | ✗ | ✓ |

categories (e.g. graphs or protein sequences).

We evaluate SMOOTHIE on one unconditional and four sequence-to-sequence generation tasks and show that it outperforms existing diffusion-based approaches. Ablation studies further demonstrate that our method enables effective control over the trade-off between fluency and diversity of the generated text.

The main contributions of our work are as follows:

- We propose a novel text diffusion framework that simultaneously respects the discrete nature of text and progressively removes semantic information from token representations during the forward process.

- We demonstrate the superior empirical performance of our approach other previous methods across multiple text generation tasks.

- We propose using a sampling method that, unlike the DDPM solver (Ho et al., 2020), ignores the previous hidden state when computing the updated hidden state. This method has been shown to consistently outperform DDPM on all sequence-to-sequence tasks used for evaluation.

## 2. Problem Statement and Background

**Problem Statement** In this work, we develop a model for both unconditional and sequence-to-sequence generation

tasks. In all cases, the objective is to generate a target sequence $\mathbf{w}^y = w_1^y, \ldots, w_m^y$. For sequence-to-sequence generation, the model additionally conditions on a source sequence $\mathbf{w}^x = w_1^x, \ldots, w_n^x$. We assume access to parallel datasets, where each source sequence is paired with its corresponding target sequence.

**Gaussian Diffusion Model** The diffusion process is defined in terms of a forward (noising) and a reverse (denoising) processes. Given an initial data point sampled from the data distribution, $\mathbf{x}_0 \sim p_{\text{data}}$, the forward process generates a sequence of progressively noisier latent variables $\mathbf{x}_1, \ldots, \mathbf{x}_T$. Each step in this sequence is defined by the transition $\mathbf{x}_t \sim q(\mathbf{x}_t \mid \mathbf{x}_{t-1}) = \mathcal{N}(\sqrt{\alpha_t}\mathbf{x}_{t-1}, \sqrt{1-\alpha_t}, \varepsilon)$, where the parameter $\alpha_t \in [0, 1)$ controls the amount of noise injected at timestep $t$. This formulation also supports a direct sampling of $\mathbf{x}_t$ from $\mathbf{x}_0$ using the marginal distribution $q(\mathbf{x}_t \mid \mathbf{x}_0) = \mathcal{N}(\sqrt{\bar{\alpha}_t}\mathbf{x}_0, \sqrt{1-\bar{\alpha}_t}, \varepsilon)$, where $\bar{\alpha}_t = \prod_{s=0}^{t} \alpha_s$ is the cumulative product of noise scales.

After the forward process is complete, a neural network $f_\theta$ is trained to reverse it by predicting the original data point $\mathbf{x}_0$ from the noisy input $\mathbf{x}_t$. During generation, the model iteratively denoises an initial sample $\mathbf{x}_T \sim \mathcal{N}(0, I)$, gradually reconstructing the data through the learned reverse process until it recovers $\mathbf{x}_0$.

**Embedding Diffusion** The most popular continuous text diffusion approaches create a latent space by mapping tokens to their embeddings (Li et al., 2022; Yuan et al., 2022;

Gong et al., 2023a). Then the Gaussian diffusion process is used to corrupt a latent. The decoding is usually performed by mapping a generated embedding to the token corresponding to the closest embedding.

**Simplex Diffusion**   SSD-LM (Han et al., 2023) and TESS (Karimi Mahabadi et al., 2024) propose a simplex diffusion model. They map each token $w$ to a $k$-logit simplex $\mathbf{s}^w \in \{\pm k\}^V$, where $V$ is the size of the vocabulary and

$$\mathbf{s}_{(i)}^w = \begin{cases} +k, & i = w \\ -k, & \text{otherwise} \end{cases} \tag{1}$$

Then the latent is represented as a sequence $\mathbf{S}_0 = (\mathbf{s}^{w_1^y}, \ldots, \mathbf{s}^{w_m^y})$. Corruption is performed with the Gaussian diffusion process with noise variance multiplied by $k^2$ ($k = 5$ by default), $\mathbf{S}_t = \sqrt{\bar{\alpha}_t}\mathbf{S}_0 + k\sqrt{1 - \bar{\alpha}_t}\varepsilon$. The model input is calculated by first producing a probability simplex over vocabulary, $\mathbf{p}_t = \text{softmax}(\mathbf{S}_t)$, and then averaging token embeddings with obtained weights, $\mathbf{p}_t\mathbf{E}$, where $\mathbf{E}$ is a matrix of token embeddings.

## 3. Related work

Since the initial attempt to apply diffusion models to text generation (Hoogeboom et al., 2021), numerous studies have explored ways to better align the diffusion process with the specifics of textual data. D3PM (Austin et al., 2021) tried exploiting the semantic property of tokens by applying a discrete diffusion process that replaces tokens with semantically similar alternatives with higher probability. However, their experiments showed that simple token masking approach produces better empirical results.

Diffusion-LM (Li et al., 2022) proposed applying Gaussian diffusion in the continuous latent space of token embeddings, while TEncDM (Shabalin et al., 2025) further demonstrated that context-dependent embeddings provide a more suitable latent space for diffusion. Despite achieving strong generation quality, the downside of these methods is the requirement of an additional latent decoding step.

DiffuSeq-v2 (Gong et al., 2023b) attempted to bridge the gap between discrete and continuous diffusion models by combining masking with Gaussian noise during the noising process. Another research direction (Han et al., 2023; Karimi Mahabadi et al., 2024) focuses on mapping tokens to almost-one-hot simplex representations over the vocabulary and introducing Gaussian noise directly into this space. While this approach does not account for token semantics during noising, it preserves the discrete structure of text.

Our work is inspired by a different line of research developed in the image domain (Rissanen et al., 2023; Hoogeboom & Salimans, 2023), where semantic information is gradually removed by smoothing pixel values according to the heat dissipation principle. This approach, however, cannot be directly applied to text due to its inherently discrete nature.

## 4. Smoothing Diffusion

In this section, we introduce SMOOTHIE, a smoothing text diffusion model that incorporates both the discrete nature of text and the semantic relationships between tokens into the diffusion process. We will first derive the diffusion process for unconditional generation and then extend it to conditional generation. We provide an intuitive illustration of our approach, along with pseudo-code for the training and sampling procedures, in Fig. 1, Alg. 1, and Alg. 2, respectively.

### 4.1. Forward Diffusion Process

Let $V$ denote the vocabulary size, and let $\mathbf{E} \in \mathbb{R}^{V \times d}$ be a fixed embedding matrix, where each row corresponds to a $d$-dimensional token embedding. To construct a latent space suitable for diffusion, we represent each token $w_i^y$ in a target sequence $\mathbf{w}^y$ with a vector of negative squared Euclidean distances between an embedding of token $w_i^y$ and embeddings of all tokens in the vocabulary:

$$\mathbf{D}_0 = \mathbf{D}_0(\mathbf{E}_{\mathbf{w}^y}) = \left\{ -\frac{\|\mathbf{E}_{w_i^y} - \mathbf{E}_j\|^2}{2} \right\}_{i,j=1}^{m,V} \tag{2}$$

Here, $m$ is the sequence length, $\mathbf{E}_{w_i^y}$ is the embedding of the i-th token in the sequence, and $\mathbf{E}_j$ is the embedding of the j-th vocabulary token. To generate a trajectory of progressively noisier latents, we define a non-Markovian forward, or noising process:

$$q(\mathbf{D}_{1:T}|\mathbf{D}_0) = \prod_{t=1}^{T} q(\mathbf{D}_t|\mathbf{D}_0) = \prod_{t=1}^{T} \mathcal{N}\left(\mathbf{D}_t \left| \frac{1}{\sigma_t^2}\mathbf{D}_0, \delta^2 I \right.\right) \tag{3}$$

The noise scheduler $\sigma_t$ ($1 < \sigma_1 < \cdots < \sigma_T$) controls the amount of noise added at each timestep. The hyperparameter $\delta$ controls the stochasticity of the diffusion process and makes it non-deterministic. Following Rissanen et al. (2023), we keep $\delta$ independent of the timestep $t$.

To construct the model input, we convert $\mathbf{D}_t$ into a probability distribution over the vocabulary using the softmax function: $\mathbf{p}_t = \text{softmax}(\mathbf{D}_t)$. In this formulation, each token is represented by the weights of Nadaraya-Watson kernel estimator applied over all embeddings in the vocabulary with Gaussian kernel whose bandwidth is defined by $\sigma_t$. The choice of a Gaussian kernel is motivated by the Euclidean semantic space hypothesis (Hashimoto et al., 2016), which assumes that semantic similarity correlates with Euclidean proximity in embedding space. As a result, as $\sigma_t$

increases, the probability mass initially centered in a single token gradually distributes between all other tokens, starting with the most semantically similar and ending with the most distant ones (see Fig. 1 (c)). We show an example of this behavior in Appendix L.

Note that our approach can be viewed as a generalization of a simplex-based diffusion (Han et al., 2023; Karimi Mahabadi et al., 2024). In particular, by replacing our Euclidean distance with trivial metric, we get the latent space formulation defined in Eq. 1, which ignores the semantic relationships between tokens. We prove this statement in Appendix A. In Section 5, we show that incorporating semantic similarity into the diffusion process is crucial for achieving better performance.

### 4.2. Reverse Diffusion Process

The reverse, or denoising process, starts with a sample from prior distribution $p(\mathbf{D}_T)$ and ends with the denoised data sample $\mathbf{D}_0$. We define it as a Markov chain with Gaussian distributions:

$$p_\theta(\mathbf{D}_{0:T}) = p(\mathbf{D}_T) \prod_{t=1}^{T} p_\theta(\mathbf{D}_{t-1}|\mathbf{D}_t) \qquad (4)$$

$$= p(\mathbf{D}_T) \prod_{t=1}^{T} \mathcal{N}\big(\mathbf{D}_{t-1}|\mu_\theta(\mathbf{p}_t, t), \tilde{\delta}^2 I\big), \qquad (5)$$

where $\theta$ are trainable model parameters and $\tilde{\delta}^2$ is a noise variance used in the reverse process. Inspired by Rissanen et al. (2023), we allow noise variance to change between the forward and reverse processes. That permits us to explicitly control the stochasticity of the generation trajectory, which significantly affects the model performance. We discuss this effect in Section 5.1.

Our goal is to find such parameters $\theta$, that minimize the marginal negative likelihood of data samples $p_\theta(\mathbf{D}_0) = \int p_\theta(\mathbf{D}_{0:T})d\mathbf{D}_{1:T}$. We optimize the negative log-likelihood by minimizing its variational upper bound, which results in the following loss function[1]:

$$L(\theta) = \mathbb{E}_q \left[ \frac{1}{2\tilde{\delta}^2} \left\| \frac{1}{\sigma_t^2}\mathbf{D}_0 - \mu_\theta(\mathbf{p}_t, t) \right\|^2 \right] \qquad (6)$$

This implies that the most direct parameterization of $\mu_\theta$ is a model that predicts $\mathbf{D}_0/\sigma_t^2$, corresponding to the posterior mean of the forward process. For practical reasons, we instead parameterize $\mu_\theta$ as $g_\theta/\sigma_t^2$ which ensures that all model outputs are scaled to have the same variance across timesteps. Also, following Ho et al. (2020), we simplify the objective by removing the scaling coefficient $2\tilde{\delta}^2\sigma_t^4$.

$$L_{\mathbf{D}}(\theta) = \mathbb{E}_{\mathbf{w}^y, t, \mathbf{p}_t} \left[ \|\mathbf{D}_0(\mathbf{E}_{\mathbf{w}^y}) - g_\theta(\mathbf{p}_t, t)\|^2 \right] \qquad (7)$$

[1]For complete derivation of the loss function see Appendix B.

---

**Algorithm 1** Training

**Input:** $\delta, \mathbf{w}^x, \mathbf{w}^y, t \sim \mathcal{U}(1, T), \varepsilon \sim \mathcal{N}(0, I)$
Compute $\mathbf{D}_0$ with Eq. 2
Compute $\mathbf{D}_t = \mathbf{D}_t/\sigma_t^2 + \delta\varepsilon$
Compute $\mathbf{p}_t = \text{softmax}(\mathbf{D}_t)$
Minimize $\|\mathbf{E}_{\mathbf{w}^y} - f_\theta(\mathbf{p}_t, t, \mathbf{w}^x)\|^2$

---

**Algorithm 2** Sampling

**Input:** Source text $\mathbf{w}^x$, model $f_\theta$, noise std $\tilde{\delta}$
Sample $\mathbf{D}_T \sim \mathcal{N}(0, \tilde{\delta}^2 I)$
**for** $t$ in $\{T, \ldots, 1\}$ **do**
  Compute $\mathbf{p}_t = \text{softmax}(\mathbf{D}_t)$
  Compute $\mathbf{D}_{t-1}$ with Eq. 9
**end for**
Decode tokens $\hat{\mathbf{w}}^y = \text{argmax}(\mathbf{D}_0)$

---

However, this loss function is challenging to optimize due to the high variance and dimensionality of $\mathbf{D}_0$. To address this issue, we introduce the following theorem:

**Theorem 4.1.** *Let $g^*(\mathbf{p}_t, t)$ be an optimal prediction for Eq. 7. Then $g^*(\mathbf{p}_t, t) = \mathbf{D}_0(f^*(\mathbf{p}_t, t)) + C$, where $C$ is a constant that does not depend on $f^*(\mathbf{p}_t, t)$ and $f^*(\mathbf{p}_t, t)$ is an optimal prediction for Eq. 8*

$$L_{\mathbf{E}}(\theta) = \mathbb{E}_{\mathbf{w}^y, t, \mathbf{p}_t} \left[ \|\mathbf{E}_{\mathbf{w}^y} - f_\theta(\mathbf{p}_t, t)\|^2 \right] \qquad (8)$$

We train the model $f_\theta$ by minimizing Eq. 8. During the sampling, we start from the random noise $\mathbf{D}_T \sim \mathcal{N}(0, \tilde{\delta}^2 I)$ and iteratively update it using the following scheme to get a clean sample:

$$\mathbf{D}_{t-1} = \frac{1}{\sigma_{t-1}^2}\mathbf{D}_0(f_\theta(\mathbf{p}_t, t)) + \tilde{\delta}\varepsilon, \qquad (9)$$

We emphasize that although the correct sampling scheme suggests using $g_\theta(\mathbf{p}_t, t)$ instead of $\mathbf{D}_0(f_\theta(\mathbf{p}_t, t))$, by the Theorem 4.1 both options are equivalent. This is because the resulting updates of these two schemes differ by a constant shift and the model on each step takes $\mathbf{p}_t = \text{softmax}(\mathbf{D}_t)$ as input, which is invariant to shifts of $\mathbf{D}_t$. The proof of Theorem 4.1 is provided in Appendix C.

Related methods such as SSD-LM (Han et al., 2023) and TESS (Karimi Mahabadi et al., 2024) employ cross-entropy loss during training. While our method is also compatible with this loss, in our experiments it led to inferior performance and faster overfitting. Therefore, we chose to rely on the MSE objective.

### 4.3. Noise Scheduler

The noise scheduler plays a crucial role in the diffusion process by controlling the rate at which the signal decays

over time. Following the observation that text diffusion models benefit from adding more noise at the early stages of the forward process (Shabalin et al., 2025), we define our noise schedule as follows:

$$\sigma_t = (\sigma_{\max} - \sigma_{\min}) \frac{2}{\pi} \arctan\left(\frac{1}{d}\sqrt{\frac{t}{T - t + \epsilon}}\right) + \sigma_{\min} \tag{10}$$

Here, $t \in [0, T]$, $\sigma_{\min}$ and $\sigma_{\max}$ set the minumum and maximum bandwidth respectively, $d$ controls the rate of noise accumulation, and $\epsilon$ is a small constant added to prevent division by zero. Throughout our experiments, we use $\sigma_{\min} = 1.5$, $\sigma_{\max} = 200$ and $d \in \{5, 7\}$ to achieve a linear increase in model entropy with increasing $t$ (Dieleman et al., 2022). Also, we set $\delta = 1$ during training. We discuss the noise scheduler ablation in Appendix H.

### 4.4. Self-conditioning

Following previous works (Dieleman et al., 2022; Yuan et al., 2022; Shabalin et al., 2025), we employ *self-conditioning* (Chen et al., 2023) to our model. During training, with 50% probability the model is fed with self-condition set to zero: $\hat{\mathbf{x}}_0^t = f_\theta(\mathbf{p}_t, \mathbf{0}, t)$. Otherwise the model receives its previous prediction as an input: $\hat{\mathbf{x}}_0^t = f_\theta(\mathbf{p}_t, \mathrm{SG}(\bar{\mathbf{x}}_0^t), t)$, where $\bar{\mathbf{x}}_0^t = f_\theta(\mathbf{p}_t, \mathbf{0}, t)$ and SG is the stop-gradient function that prevent gradients from flowing through $\bar{\mathbf{x}}_0^t$. During the generation stage, the first prediction is made with self-condition set to zero and at all subsequent steps the predictions are performed as $\hat{\mathbf{x}}_0^t = f_\theta(\mathbf{p}_t, \hat{\mathbf{x}}_0^{t+1}, t)$. We demonstrate the impact of self-conditioning in Appendix G.

### 4.5. Sequence Length

As diffusion models operate over fixed-length sequences, we pad all shorter sequences using a special padding token, which the model is trained to predict. In the end of generation padding tokens are discarded. To limit computational overhead, we set the maximum sequence length for each dataset to approximately the 99th percentile of training set sequence lengths. The exact values used for each dataset are provided in the Appendix J.

## 5. Experiments

**Implementation Details** In all experiments, we use a pretrained embedding matrix, $\mathbf{E}$, from the BERT (Devlin et al., 2019) model[2]. We normalize this matrix to have a zero mean and a unit variance and keep it fixed throughout training. Although the model receives the soft token distribution $\mathbf{p}_t$ as input, it does not operate directly on this distribution.

---

[2]We consider other embedding options in Appendix F

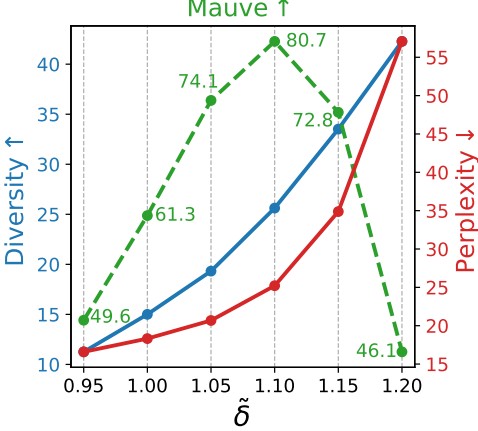

*Figure 2.* Unconditional generation for $\delta = 1$ and varying $\tilde{\delta}$.

Instead, we compute a weighted average of the token embeddings, $\mathbf{p}_t \mathbf{E}$, which yields a lower-dimensional, more tractable representation for the model to process.

Our model's architecture is based on the design proposed in Shabalin et al. (2025), consisting of Transformer decoder layers (Vaswani et al., 2017) augmented with UNet-style skip connections. Specifically, the output of the first layer is added to the input of the last, the second to the second-last, and so on. The full model has approximately 100M parameters. For conditional generation, we modify the model to accept an input sequence $\mathbf{w}^x$, which is processed by an additional 6-layer Transformer encoder. The encoder output is integrated into the decoder through cross-attention mechanisms. For timestep conditioning, we adopt the approach from Gong et al. (2023a), plugging learned timestep embeddings into each Transformer block akin to positional embeddings. The complete set of hyperparameters used for training and evaluation is provided in Appendix J.

### 5.1. The Importance of $\tilde{\delta}$

Before presenting results on seq-to-seq generation tasks, we highlight the importance of the hyperparameter $\tilde{\delta}$, which controls the stochasticity of the denoising process. To illustrate its impact, we evaluate generation quality on an *unconditional* generation task using different values of $\tilde{\delta}$. Specifically, we use the **ROCStories** dataset and assess performance using three metrics: generative **perplexity** (to estimate average text quality), **diversity** (to measure lexical variety) (Su et al., 2022), and the **MAUVE Score** (Pillutla et al., 2021) (to evaluate the overall similarity of generated texts to the reference distribution). When calculating MAUVE, we generate 1,000 texts five times with different seeds and compare them with 1,000 randomly sampled reference texts. We then average the results.

Figure 2 shows the results for a model trained with $\delta = 1$. We observe that lower values of $\tilde{\delta}$ lead to better perplexity scores but lower diversity. In other words, reduced stochasticity improves the quality of individual texts but decreases their uniqueness. This trade-off is actually desirable for sequence-to-sequence tasks, where diversity typically arises naturally from the varying input conditions. In Appendix E, we justify this insight by grid-searching the best $\tilde{\delta}$ value. As a result, we set $\tilde{\delta} = 0.1$ for all sequence-to-sequence experiments.

In contrast, for unconditional generation, the optimal value of $\tilde{\delta}$ is slightly higher than the one used during training, as indicated by the MAUVE Score. At this point, the generated texts exhibit sufficient diversity while maintaining acceptable perplexity. These findings show that $\tilde{\delta}$ has a strong influence on the generation process and should be tuned carefully depending on the target task.

**Datasets** In addition to the unconditional generation on **ROCStories** dataset, we evaluate SMOOTHIE on four sequence-to-sequence datasets of varying difficulty. For *paraphrase generation*, we use the Quora Question Pairs (**QQP**) dataset (Chen et al., 2017), which contains 147K pairs of semantically equivalent questions. For *question generation*, we adopt the **Quasar-T** dataset (Dhingra et al., 2017), processed by Gong et al. (2023a), resulting in 119K document-question pairs. For *summarization*, we use the **XSum** dataset (Narayan et al., 2018), comprising 204K BBC articles and their corresponding summaries. For *detoxification*, we use **ParaDetox** (Logacheva et al., 2022) dataset with 19,766 pairs of toxic and neutral comments. More detailed dataset information is provided in the Appendix K.

**Metrics** Following the evaluation protocol from Gong et al. (2023a); Karimi Mahabadi et al. (2024), we employ a combination of n-gram-based, diversity and semantic similarity metrics. Specifically, we report **BLEU** (Papineni et al., 2002) and **ROUGE-1/2/L** (Lin, 2004) scores to measure lexical overlap between generated and reference texts, and **BERTScore (BS)** (Zhang et al., 2020) to assess semantic similarity. For BERTScore, we use the `microsoft/deberta-xlarge-mnli` model to ensure consistency with previous studies (Yuan et al., 2022; Karimi Mahabadi et al., 2024).

To evaluate the diversity of generated texts, we compute n-gram diversity (Deshpande et al., 2019), which reports the fraction of unique unigrams (**Div-1**) and 4-grams (**Div-4**) in a text. Additionally, for the text detoxification task, we measure **J-Score** (Krishna et al., 2020), which comprises text fluency, style accuracy, and content preservation.

**Baselines** We compare SMOOTHIE against several diffusion-based and autoregressive baselines, all with ap-

*Table 2.* Generation time and peak memory consumption for SMOOTHIE, embedding- and simplex-based diffusion.

| Method | Generation time (s) | Memory (MB) |
|---|---|---|
| Embedding | 1.642 | 593.3 |
| Simplex | 2.897 | 678.4 |
| SMOOTHIE | 2.897 | 593.3 |

proximately 100M parameters and trained from scratch on each dataset. The diffusion-based baselines include DiffuSeq (Gong et al., 2023a), SeqDiffuSeq (Yuan et al., 2022), AR-Diffusion (Wu et al., 2023), and GENIE (Lin et al., 2023), SSD-LM (Han et al., 2023), TESS (Karimi Mahabadi et al., 2024), and TEncDM (Shabalin et al., 2025). We also compare against MDLM (Sahoo et al., 2024), an established masked diffusion model that we trained for sequence-to-sequence tasks using the provided code. For autoregressive baselines, we evaluate BART (Lewis et al., 2020), GPT-2 (Radford et al., 2019), GPVAE-T5 (Du et al., 2022), FLAN-T5 (Chung et al., 2024), and a standard Transformer (Vaswani et al., 2017). TESS approach uses pre-trained RoBERTa (Liu et al., 2019) to initialize its diffusion model. For a fair comparison, we only compare to the model trained from random initialization.

Additionally, we conduct a rigorous comparison of our proposed distance-based latent space with two previously explored alternatives: the embedding space (Yuan et al., 2022; Gong et al., 2023a) (Embedding* in experiments) and the simplex space (Han et al., 2023; Karimi Mahabadi et al., 2024) (Simplex* in experiments). To ensure a fair evaluation, we train all diffusion models under identical conditions, keeping the architecture, training hyperparameters, and decoding strategy fixed. The only variables are the latent space and its associated noise schedule. For embedding-based diffusion, we use the noise scheduler from Shabalin et al. (2025), while for simplex-based diffusion, we adopt the scheduler from Han et al. (2023). In all three cases, sampling is performed using a procedure defined in the respective latent space, following the formulation in Eq. 9. SMOOTHIE and the embedding-based diffusion model are trained using MSE loss, while the simplex-based diffusion is trained using cross-entropy loss because it is not suitable for predicting continuous embeddings.

**Generation Speed** During generation, SMOOTHIE requires the calculation of pairwise distances between the predicted embeddings and all the embeddings in the vocabulary. This operation has a complexity of $\mathcal{O}($batch size $\times$ seq len $\times d \times V)$, which is the same as that of the linear head used in simplex diffusion (Karimi Mahabadi et al., 2024; Lee et al., 2026) or discrete diffusion models (Austin et al., 2021; Sahoo et al., 2024; Deschenaux et al., 2026) to predict tokens at each step. However, embedding-based diffusion

*Table 3.* Results on XSum (left) and Quasar-T (right) datasets. † denotes autoregressive models, △ denotes the results reproduced with original code, ⋆ denotes our implementations. The best-performing *diffusion* results are highlighted in **bold**, the second-best are underlined.

| | XSum | |
|---|---|---|
| **Method** | **BS ↑** | **R-1/2/L ↑** |
| GPT-2[†△] | 69.0 | 28.3/8.2/21.8 |
| Transformer[†] | — | 30.5/10.4/24.2 |
| FLAN-T5[†] | 72.7 | 34.6/12.9/27.2 |
| MDLM[△] | 62.1 | 27.9/7.7/21.1 |
| DiffuSeq | 46.8 | 18.9/1.3/13.6 |
| SeqDiffuSeq[△] | 61.8 | 28.6/6.7/21.3 |
| AR-Diffusion | — | 31.7/10.1/24.7 |
| GENIE | — | 29.3/8.3/21.9 |
| TEncDM | **69.9** | 31.9/10.7/25.3 |
| Embedding⋆ | 68.2 | 32.1/10.1/24.6 |
| Simplex⋆ | 63.8 | 29.6/8.5/23.0 |
| Smoothie⋆ (ours) | 68.8 | **33.7/11.1/26.0** |

| | Quasar-T | | | |
|---|---|---|---|---|
| **Method** | **BS ↑** | **BLEU ↑** | **R-L ↑** | **D-1/4** |
| GPT-2[†] | 60.5 | 7.4 | 27.2 | 96.0/92.2 |
| GPVAE-T5[†] | 63.1 | 12.5 | 33.9 | 93.8/72.8 |
| BART[†] | 66.2 | 17.4 | 38.8 | 98.2/61.7 |
| MDLM[△] | 60.7 | 17.5 | 33.6 | 91.0/**64.2** |
| DiffuSeq | 59.4 | 15.8 | — | 91.1/— |
| SeqDiffuSeq | 61.4 | 17.2 | — | 92.7/— |
| SSD-LM | 62.8 | 14.1 | **38.5** | 94.5/56.9 |
| TESS (random) | 60.8 | 19.0 | 36.1 | **96.1**/62.4 |
| Embedding⋆ | 62.0 | 18.9 | 35.2 | 92.4/61.2 |
| Simplex⋆ | 63.0 | 19.3 | 36.9 | 93.0/63.8 |
| Smoothie⋆ (ours) | **63.1** | **19.9** | 36.5 | 92.8/63.3 |

does not have such operation and thus generates text faster[3].

We measure the difference in generation time and memory usage between the three approaches and present the results in Table 2. We performed 100 generation steps with a batch size of 32 and a sequence length of 80, reporting the total generation time and peak memory consumption. We observe that simplex diffusion and Smoothie have the same speed, while embedding diffusion is $1.75\times$ faster. To compensate for this, we increase the number of generation steps by $1.75\times$ for embedding diffusion compared to Smoothie in all our experiments.

## 5.2. Empirical Results

We now present a quantitative comparison of Smoothie against a range of generative models. Wherever possible, we adopt reported results from prior works (Lovelace et al., 2023; Wu et al., 2023; Karimi Mahabadi et al., 2024; Meshchaninov et al., 2025). When certain metric values are unavailable, we reproduce the corresponding methods using the original implementations. For consistency, we re-implement and train the embedding- and simplex-based diffusion baselines within our framework.

We show the results on XSum and Quasar-T dataset in Table 3, and on ROCStories, QQP, and ParaDetox in Table 4. In Appendix D, we also evaluate Smoothie of OpenWebText (Gokaslan et al., 2019). Overall, Smoothie consistently outperforms other text diffusion approaches, as well as diffusion methods based on embedding- and simplex-based latent spaces achieving quality comparable to that of autoregressive models.

Notably, embedding-based diffusion performs better than

simplex-based diffusion on all datasets except Quasar-T. This difference can be attributed to the fact that simplex-based diffusion does not incorporate semantic information into the noising process, making it inherently more chaotic. Nevertheless, when combined with our proposed architecture, simplex-based diffusion surpasses the TESS approach, which employs the same diffusion process and a training pipeline, differing only in the architecture design. This highlights that selecting an appropriate model architecture is as critical as choosing the diffusion space.

The most pronounced improvement in generation quality is observed on the ROCStories dataset. By tuning the $\tilde{\delta}$ parameter (Section 5.1), Smoothie effectively balances diversity and coherence, achieving the highest MAUVE score and nearly matching the quality of GPT-2.

## 5.3. Amount of Denoising Steps

Table 5 presents the relationship between the number of denoising steps and the generation quality of Smoothie in terms of J-Score for ParaDetox and BERTScore for other datasets. We observe that for all datasets except ParaDetox, the quality does not change much regardless of the number of steps. Nevertheless, for XSum the performance improves as the number of steps increases until we reach 200 steps, after which the quality drops. This can be explained by the impact of self-conditioning, which lead to a mismatch between train and generation trajectory for larger amount of steps (Shabalin et al., 2025). Overall, the results align with the observation made in the TESS paper (Karimi Mahabadi et al., 2024), which suggests that the optimal number of denoising steps correlates with the complexity of the task.

---

[3]Some papers use a clamping trick (Li et al., 2022) that results in the same overall computational complexity as Smoothie.

*Table 4.* Text generation results on ROCStories, QQP and ParaDetox datasets. † denotes autoregressive models, △ denotes the results reproduced with original code, ⋆ denotes our implementations. The best-performing *diffusion* results are highlighted in **bold**, the second-best are underlined.

| Method | ROCStories | | | QQP | | | | ParaDetox | |
| | MAUVE ↑ | PPL ↓ | Div ↑ | BS ↑ | BLEU ↑ | R-L ↑ | D-1/4 ↑ | BLEU ↑ | J-Score ↑ |
|---|---|---|---|---|---|---|---|---|---|
| GPT-2† | 78.9 | 20.5 | 25.2 | 82.5 | 19.8 | 52.1 | 98.0/62.5 | 67.7 | 60.4 |
| GPVAE-T5† | — | — | — | 84.7 | 24.1 | 58.9 | 96.9/61.7 | — | — |
| BART† | — | — | — | 85.7 | 30.4 | 61.4 | 98.8/61.0 | — | — |
| MDLM△ | 63.9 | 58.1 | **35.1** | 76.3 | 21.5 | 46.2 | 96.2/64.4 | 61.5 | 41.4 |
| DiffuSeq | 8.6 | 50.5 | 12.4 | 79.5 | 18.5 | — | 97.6/— | 67.9 | 47.5 |
| SeqDiffuSeq | 10.3 | 29.3 | 13.7 | 82.9 | 23.3 | — | 98.1/— | 68.8 | 48.6 |
| AR-Diffsion△ | 6.6 | 41.8 | 10.1 | 80.1 | 19.2 | 54.9 | — | 64.7 | 46.5 |
| SSD-LM | — | — | — | 83.8 | 22.9 | 58.3 | **98.8**/57.3 | — | — |
| TEncDM | 76.2 | 29.1 | 29.5 | 83.8 | 30.7 | 57.3 | — | 61.9 | 49.6 |
| Embedding⋆ | 41.5 | 28.3 | 26.1 | 83.4 | **31.3** | 59.4 | 97.7/64.5 | 67.6 | 49.1 |
| Simplex⋆ | 15.2 | 25.3 | 12.4 | 80.6 | 26.8 | 54.9 | 96.8/**64.8** | 65.1 | 47.7 |
| Smoothie⋆ (ours) | **80.7** | **24.9** | 25.1 | **83.9** | 30.8 | **60.9** | 98.4/60.5 | **69.2** | **51.7** |

*Table 5.* The impact of changing the number of steps on generation quality. We show J-score for ParaDetox and BERTScore for the other datasets.

| Steps | XSum | Quasar-T | QQP | ParaDetox |
|---|---|---|---|---|
| 25 | 67.7 | **63.1** | **83.9** | 51.1 |
| 50 | 68.5 | **63.1** | 83.8 | 51.4 |
| 100 | 68.7 | **63.1** | 83.7 | **51.7** |
| 200 | **68.8** | **63.1** | 83.6 | 51.0 |
| 500 | 68.4 | **63.1** | 83.5 | 50.8 |

*Table 6.* Generation quality of DDPM, DDIM and our (Eq. 9) diffusion solvers on Seq2seq tasks.

| Solver | XSum | | Quasar-T | | QQP | | ParaDetox |
| | BS ↑ | R-L ↑ | BS ↑ | R-L ↑ | BS ↑ | R-L ↑ | J-Score ↑ |
|---|---|---|---|---|---|---|---|
| DDPM | 68.5 | 25.6 | 62.8 | 35.5 | 83.7 | 60.7 | 51.5 |
| DDIM | 67.4 | 24.6 | 60.3 | 31.8 | 82.0 | 57.9 | 50.2 |
| Ours | **68.8** | **26.0** | **63.1** | **36.5** | **83.9** | **60.9** | **51.7** |

## 5.4. Diffusion Solver

In all experiments, we use the solver defined in Eq. 9 rather than the commonly used DDPM (Ho et al., 2020) or DDIM (Song et al., 2021) solvers. This choice is motivated by the consistently superior empirical performance of our solver. To enable a comparison with DDPM and DDIM, we adapt the originally proposed forward process (Eq. 3) by introducing a timestep-dependent variance.

$$q(\mathbf{D}_t|\mathbf{D}_0) = \mathcal{N}\left(\mathbf{D}_t \middle| \frac{1}{\sigma_t^2}\mathbf{D}_0, \left(1 - \frac{1}{\sigma_t^4}\right)\delta^2 I\right) \quad (11)$$

A quantitative comparison between DDPM, DDIM and our solver is reported in Table 6. The results demonstrate a marginal yet consistent advantage of our solver across all evaluated tasks. We hypothesize that this behavior stems from the discrete nature of text embeddings, which leads to a highly sparse data distribution. Such sparsity may be unfavorable for both the DDPM and DDIM solvers, as they move the lantent in a predicted direction and even a small deviation in this direction can cause the latent representation to leave the data manifold, which will cause more inaccuracy in further predictions.

In contrast, our solver does not rely on the previous latent state when computing updates. Instead, it directly applies the forward process to the model's prediction. Although this strategy may appear suboptimal, it reduces the risk of drifting in incorrect directions, which is particularly important for sparse distributions. Nevertheless, a deeper theoretical understanding of the observed performance gap remains an open question and is left for future work.

## 6. Limitations

**Pre-trained Embeddings** SMOOTHIE relies on a pre-trained embedding matrix $\mathbf{E}$ from the BERT model. While this choice simplifies the training process and improves its stability, it may hold back the model's generation quality. Finetuning embeddings for a specific task should offer better results. Also, an end-to-end training approach, as used in Li et al. (2022); Gong et al. (2023a), could be applied to our method. However, training embeddings simultaneously with the diffusion model is hard to stabilize, because embeddings collapse when diffusion model is optimized only with MSE between text embeddings and model's predictions. Addition of a cross-entropy loss prevents the collapse, but might lead to the explosion of embeddings' norm, again making the task trivial for the diffusion model. For these reasons, we leave the exploration of an end-to-end training approach for future work.

**Complexity of the Distance Matrix Computation** SMOOTHIE requires computing a matrix of pairwise distances between the predicted embedding and all vocabulary

embeddings at every generation step, which slows down sampling. While several related approaches such as SSD-LM (Han et al., 2023), TESS (Karimi Mahabadi et al., 2024), and Diffusion-LM (Li et al., 2022) have the same sampling complexity, alleviating this bottleneck is an important direction for improvement.

Unfortunately, this issue cannot be easily addressed by straightforward approximations, such as restricting the computation to top-$k$ nearest embeddings. At most timesteps, nearly all vocabulary embeddings receive non-zero weights $\mathbf{p}_t$, making such approximation inaccurate. In principle, this limitation could be mitigated by clustering embeddings at each timestep and performing smoothing using cluster centroids, or by reducing the effective vocabulary size. However, both of these methods are very complex and go beyond the scope of this work.

## 7. Conclusion

In this work, we introduce SMOOTHIE, a text diffusion method that constructs its diffusion process with consideration of the discrete nature of text and the semantic relationships between tokens. To capture these properties, each token is mapped to a vector of Euclidean distances between its embedding and the embeddings of all tokens in the vocabulary. Our choice of the Euclidean distance is based on the Euclidean semantic space hypothesis (Hashimoto et al., 2016), which posits that semantic similarity correlates with Euclidean proximity in embedding space. Additionally, we use a sampling method that disregards the previous latent state when calculating the updated latent. This solver demonstrates superior empirical performance to the conventionally used DDPM.

Our method also can be applicable to other categorical domains where semantic relationships exist between categories (e.g. graphs, protein sequences). However, in such cases, a different distance metric more suited to the domain's properties may be required. We leave the exploration of this direction to future work.

Empirical results on four sequence-to-sequence tasks demonstrate that SMOOTHIE outperforms existing text diffusion methods, as well as our diffusion model framework with alternative diffusion latent spaces that do not rely on additional encoders.

## Acknowledgments

We are grateful to Ildus Sadrtdinov for his valuable insights and discussions throughout this project. The work was supported by the grant for research centers in the field of AI provided by the Ministry of Economic Development of the Russian Federation in accordance with the agreement 000000C313925P4E0002 and the agreement with HSE University №139-15-2025-009. The paper was supported in part through computational resources of HPC facilities at HSE University.

## Impact Statement

This paper presents work whose goal is to advance the field of Natural Language Processing. There are many potential societal consequences of our work, none which we feel must be specifically highlighted here.

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

## A. Relationship Between Distance-based and Simplex-based Latent Spaces

In this section, we demonstrate that our proposed *distance-based latent space* generalizes the *simplex-based latent space*. Specifically, we show that the simplex-based latent space corresponds to a special case of a distance-based latent space when equipped with a trivial metric.

SMOOTHIE maps each token $w$ to a latent vector $\mathbf{d}^w$, where each component is given by:

$$\mathbf{d}^w_{(i)} = -\frac{1}{2}\|\mathbf{E}_w - \mathbf{E}_i\|^2. \tag{12}$$

For other categorical domains, the Euclidean distance can be replaced with a more suitable metric $\rho(w, i)$, leading to:

$$\mathbf{d}^w_{(i)} = -\rho(w, i). \tag{13}$$

To relate this to simplex-based representations, consider the case where $\rho$ is the *trivial metric*:

$$\rho(w, i) = [w \neq i], \tag{14}$$

i.e., 0 when $w = i$ and 1 otherwise. Under this choice, the latent vector becomes:

$$\mathbf{d}^w_{(i)} = \begin{cases} 0, & i = w, \\ -1, & \text{otherwise}. \end{cases} \tag{15}$$

In comparison, the simplex-based latent space maps each token $w$ to a vector $\mathbf{s}^w$ in the $k$-logit simplex:

$$\mathbf{s}^w_{(i)} = \begin{cases} +k, & i = w, \\ -k, & \text{otherwise}. \end{cases} \tag{16}$$

Both SMOOTHIE and simplex diffusion apply a Gaussian diffusion process to corrupt the latent vector:

$$\mathbf{z}_t = \phi_t \mathbf{z}_0 + \gamma_t \varepsilon, \tag{17}$$

where $\mathbf{z}_0 \in \{\mathbf{d}^w, \mathbf{s}^w\}$ and $\varepsilon \sim \mathcal{N}(0, I)$. To form a model input, the corrupted vector is then transformed into a probability distribution using the softmax function:

$$p_t = \text{softmax}(\mathbf{z}_t). \tag{18}$$

Since the softmax function is invariant to uniform additive shifts, we have:

$$\text{softmax}(\phi_t \mathbf{s}^w + \gamma_t \varepsilon) = \text{softmax}(\phi_t(\mathbf{s}^w - k) + \gamma_t \varepsilon) = \text{softmax}(2k\phi_t \mathbf{d}^w + \gamma_t \varepsilon), \tag{19}$$

where the final equality follows from observing that $\mathbf{s}^w - k = 2k\mathbf{d}^w$.

This confirms that the simplex-based latent space is equivalent, up to scaling, to the distance-based latent space under the trivial metric. Hence, the simplex-based representation is a special case within the more general distance-based latent space framework.

## B. Derivation of the Loss Function

$$-\log p_\theta(\mathbf{D}_0) = -\log \int \frac{p_\theta(\mathbf{D}_{0:T})q(\mathbf{D}_{1:T}|\mathbf{D}_0)}{q(\mathbf{D}_{1:T}|\mathbf{D}_0)} d\mathbf{D}_{1:T} \leq -\mathbb{E}_q \log \frac{p_\theta(\mathbf{D}_{0:T})}{q(\mathbf{D}_{1:T}|\mathbf{D}_0)} \tag{20}$$

$$= -\mathbb{E}_q \left[ \log \frac{p_\theta(\mathbf{D}_T)}{q(\mathbf{D}_T|\mathbf{D}_0)} + \sum_{t=2}^{T} \log \frac{p_\theta(\mathbf{D}_{t-1}|\mathbf{D}_t)}{q(\mathbf{D}_{t-1}|\mathbf{D}_0)} + \log p_\theta(\mathbf{D}_0|\mathbf{D}_1) \right] \tag{21}$$

$$= \mathbb{E}_q \left[ \underbrace{D_{\text{KL}}\big[q(\mathbf{D}_T|\mathbf{D}_0)\|p(\mathbf{D}_T)\big]}_{L_T} + \sum_{t=2}^{T} \underbrace{D_{\text{KL}}\big[q(\mathbf{D}_{t-1}|\mathbf{D}_0)\|p_\theta(\mathbf{D}_{t-1}|\mathbf{D}_t)\big]}_{L_{t-1}} \underbrace{-\log p_\theta(\mathbf{D}_0|\mathbf{D}_1)}_{L_0} \right] \tag{22}$$

In this formula, $L_T$ is constant during the training, as it does not depend on any learnable parameters. Both forward and reverse processes are defined by Gaussian distributions, which allows us to compute the terms $L_0$ and $L_{t-1}$ in closed form:

$$L_0 = \mathbb{E}_q \left[ \frac{1}{2\tilde{\delta}^2} \|\mathbf{D}_0 - \mu_\theta(\mathbf{p}_1, 1)\|^2 \right] + C_0; \quad L_{t-1} = \mathbb{E}_q \left[ \frac{1}{2\tilde{\delta}^2} \left\| \frac{1}{\sigma_t^2} \mathbf{D}_0 - \mu_\theta(\mathbf{p}_t, t) \right\|^2 \right] + C_{t-1}, \tag{23}$$

where $C_0$ and $C_{t-1}$ are constants that do not depend on parameters $\theta$. This implies that the most direct parameterization of $\mu_\theta$ is a model that predicts $\mathbf{D}_0/\sigma_t^2$, corresponding to the posterior mean of the forward process. However, for practical reasons, we instead parameterize $\mu_\theta$ as $g_\theta/\sigma_t^2$ which ensures that all model outputs are scaled to have the same variance across timesteps.

$$L_{t-1} = \mathbb{E}_q \left[ \frac{1}{2\tilde{\delta}^2 \sigma_t^4} \|\mathbf{D}_0 - g_\theta(\mathbf{p}_t, t)\|^2 \right] + C_{t-1}, \tag{24}$$

Following Ho et al. (2020), we replace $L_{t-1}$ with its simplified version by removing the scaling coefficient $2\tilde{\delta}^2 \sigma_t^4$, resulting in the following loss function:

$$L_\mathbf{D}(\theta) = \mathbb{E}_{\mathbf{w}^y, t, \mathbf{p}_t} \left[ \|\mathbf{D}_0(\mathbf{E}_{\mathbf{w}^y}) - g_\theta(\mathbf{p}_t, t)\|^2 \right] \tag{25}$$

## C. Proof of Theorem 4.1

*Proof.* We begin by recalling a standard result:

**Lemma.** The minimum value of the function $\mathbb{E}_\mathbf{y} \left[ \|\mathbf{y} - \mathbf{z}\|^2 \right]$ is achieved when $\mathbf{z} = \mathbb{E}[\mathbf{y}]$.

Using this lemma, we obtain:

$$g^*(\mathbf{p}_t, t) = \mathbb{E}_{\mathbf{w}^y}[\mathbf{D}_0(\mathbf{E}_{\mathbf{w}^y})] = \mathbb{E}_{\mathbf{w}^y} \left[ -\frac{1}{2} \{ \|\mathbf{E}_{w_i^y} - \mathbf{E}_j\|^2 \}_{i,j=1}^{m,V} \right] \quad \text{and} \quad f^*(\mathbf{p}_t, t) = \mathbb{E}_{\mathbf{w}^y}[\mathbf{E}_{\mathbf{w}^y}], \tag{26}$$

where $\mathbf{w}^y \sim p(\mathbf{w}^y \mid \mathbf{p}_t)$. Since both $g^*(\mathbf{p}_t, t)$ and $f^*(\mathbf{p}_t, t)$ are matrices, without loss of generality we will prove this statement for an arbitrary row $i$ and column $j$. For brevity, we will define $u = \mathbf{E}_{w_i^y}$ and $v = \mathbf{E}_j$. Then, we need to show that

$$\mathbb{E}_u \left[ -\frac{1}{2} \|u - v\|^2 \right] = -\frac{1}{2} \|\mathbb{E}[u] - v\|^2 + C \tag{27}$$

Expanding both sides:

$$\mathbb{E}_u \left[ \|u - v\|^2 \right] = \mathbb{E}[\|u\|^2] - 2v^\top \mathbb{E}[u] + \|v\|^2$$
$$\|\mathbb{E}[u] - v\|^2 = \|\mathbb{E}[u]\|^2 - 2v^\top \mathbb{E}[u] + \|v\|^2$$

Subtracting:

$$\mathbb{E}[\|u\|^2] - \|\mathbb{E}[u]\|^2 = \sum_{k=1}^d \mathrm{Var}(u_k) = C$$

Thus,

$$\mathbb{E}_u \left[ -\frac{1}{2} \|u - v\|^2 \right] = -\frac{1}{2} \|\mathbb{E}[u] - v\|^2 - \underbrace{\frac{1}{2}C}_{\text{constant}},$$

where $C$ is a constant independent of $\mathbb{E}[u]$.

Since this holds for all $(i, j)$, the matrix identity holds:

$$g^*(\mathbf{p}_t, t) = \mathbf{D}_0(f^*(\mathbf{p}_t, t)) + \mathbf{C}$$

$\square$

## D. OpenWebText results

In this section, we provide the numerical results for SMOOTHIE, embedding-based diffusion and GIDD, a strong masked diffusion model (von Rütte et al., 2025) on the large scale OpenWebText dataset (Gokaslan et al., 2019) with the sequence length on 512 tokens. We use DDPM solver for embedding diffusion, and for SMOOTHIE we perform 40% of first steps with DDPM and switch to the solver in Eq. 9 for the remaining steps. We run SMOOTHIE for 100 steps, and, to align the methods in terms of computational complexity, we use $1.75\times$ more steps for embedding diffusion. Table 7 shows that SMOOTHIE significantly outperforms GIDD and marginally surpasses embedding diffusion by MAUVE metric.

*Table 7.* Text generation results on the OpenWebText dataset. $\triangle$ denotes the results reproduced with original code, $\star$ denotes our implementations.

| Method | MAUVE ↑ | PPL ↓ | Div ↑ |
|---|---|---|---|
| GIDD$^{\triangle}$ (von Rütte et al., 2025) | 28.6 | 228.3 | **58.8** |
| Embedding$^{\star}$ | 55.7 | 51.9 | 21.2 |
| SMOOTHIE$^{\star}$ ($\tilde{\delta} = 0.96$) | 58.9 | **48.2** | 19.8 |
| SMOOTHIE$^{\star}$ ($\tilde{\delta} = 1.02$) | **59.9** | 62.2 | 23.4 |

## E. An Impact of $\tilde{\delta}$ on Seq2seq Tasks

In this section, we measure how the quality of sequence-to-sequence generation changes when the value of $\tilde{\delta}$ varies. For this experiment, we consider values in the range of $\tilde{\delta} \in \{0.1, 0.25, 0.5, 0.75, 1\}$ and set the number of generation steps to 100. Table 8 reports J-Score for ParaDetox and BERTScore for all other datasets. Although the difference in quality for different $\tilde{\delta}$ is not as significant as for the unconditional generation, it can be seen that lower values of $\tilde{\delta}$ produce better quality overall. Following these results, we set $\tilde{\delta} = 0.1$ for all datasets.

*Table 8.* The impact of $\tilde{\delta}$ value on generation quality. We show J-Score for ParaDetox and BERTScore for the other datasets.

| $\tilde{\delta}$ | XSum | Quasar-T | QQP | ParaDetox |
|---|---|---|---|---|
| 0.1 | **68.8** | **63.1** | **83.7** | **51.7** |
| 0.25 | 68.7 | **63.1** | **83.7** | 51.4 |
| 0.5 | 68.7 | **63.1** | **83.7** | 51.3 |
| 0.75 | 68.6 | **63.1** | 83.6 | 50.9 |
| 1 | 68.2 | **63.1** | 83.4 | 50.9 |

## F. Embeddings Ablation

Throughout this work, we utilize BERT embedding matrix to represent text tokens without additional comments. We find it important to evaluate the robustness of SMOOTHIE to other choices of embeddings. Therefore, we demonstrate how model performance changes on the ROCStories dataset when embeddings are changed. We choose two alternatives with the same hidden size: GPT-2 (Radford et al., 2019) embeddings with the vocabulary size of 50k and GloVe (Pennington et al., 2014) embeddings trained manually on Wikipedia dataset for BPE tokens with the vocabulary size of 10k. In the Table 9 we show the results of the ablation.

*Table 9.* The generation quality of SMOOTHIE trained with different embedding types on ROCStories dataset.

| Embeddings | MAUVE ↑ | PPL ↓ | Div ↑ |
|---|---|---|---|
| BERT (default) | 80.7 | 24.9 | 25.1 |
| GPT-2 | 64.4 | 23.1 | 25.0 |
| GloVe | 36.8 | 36.4 | 24.6 |

In terms of perplexity and diversity, GPT-2 embeddings perform similarly to BERT, with the exception of MAUVE. However, these results are still better than those of most other methods (see Table 4). Interestingly, we found out that the optimal value of $\tilde{\delta}$ for GPT2 embeddings is lower than for BERT embeddings (1.03 vs 1.1). Most probably, this is because diversity increases naturally with the increase of the vocabulary size and the need to increase it artificially disappears. GloVe embeddings are worse than the ones extracted from a language model. Therefore, a significant drop in quality is not surprising. We can conclude that embeddings is an important component of the framework and the quality of the model does depend on the quality of embeddings. However, the method allows for flexibility in the choice of embeddings, which improves its applicability.

*Table 10.* An impact of the parameter $d$ in noise scheduler on the generation quality on the ROCStories dataset.

| | MAUVE ↑ | PPL ↓ | Div ↑ |
|---|---|---|---|
| $d = 4$ | 74.2 | 24.4 | 24.5 |
| $d = 5$ | 80.7 | 24.9 | 25.1 |
| $d = 6$ | 76.5 | 26.7 | 27.7 |
| $d = 7$ | 68.9 | 24.6 | 26.7 |

## G. Self-conditioning

Previous studies have shown that self-conditioning significantly improves the quality of text diffusion models (Yuan et al., 2022; Shabalin et al., 2025; Karimi Mahabadi et al., 2024; Dieleman et al., 2022). In this section, we compare the performance of SMOOTHIE, as well as of embedding- and simplex-based diffusion models, with and without self-

*Table 11.* Impact of self-conditioning on the generation performance on XSum, Quasar-T and QQP datasets.

| Method | XSum | | Quasar-T | | | QQP | | |
|---|---|---|---|---|---|---|---|---|
| | **BS ↑** | **R-L ↑** | **BS ↑** | **BLEU ↑** | **R-L ↑** | **BS ↑** | **BLEU ↑** | **R-L ↑** |
| Embedding | 68.2 | 24.6 | 62.0 | 18.9 | 35.2 | 83.5 | 31.6 | 59.6 |
| w/o SC | 65.2 | 23.6 | 62.9 | 19.5 | 36.0 | 81.7 | 27.7 | 57.4 |
| Simplex | 63.8 | 23.0 | 63.0 | 19.3 | 36.9 | 81.2 | 27.3 | 55.0 |
| w/o SC | 61.2 | 21.5 | 62.5 | 19.4 | 36.4 | 80.0 | 25.9 | 54.1 |
| SMOOTHIE | 68.8 | 26.0 | 63.0 | 19.0 | 35.8 | 83.9 | 30.8 | 60.9 |
| w/o SC | 67.5 | 25.4 | 61.9 | 19.0 | 35.7 | 83.2 | 29.4 | 59.9 |

conditioning. The results on the XSum, Quasar-T, and QQP datasets are reported in Table 11. Although performance gains vary across models and datasets, self-conditioning generally improves quality, which confirms the previous observations.

## H. Noise Scheduler Ablation

In this work, we use a special *arctan* noise scheduler for SMOOTHIE to make sure that the model entropy grows linearly with $t$ (Dieleman et al., 2022). In this section, we perform an ablation study for the proposed noise scheduler by evaluating different values of $d$. In Table 10, we show the numerical performance on the ROCStories dataset. For each $d$ we chose the best $\tilde{\delta}$ based on MAUVE. Smaller $d$ values correspond to more aggressive corruption. The results suggest that while the difference is marginal, SMOOTHIE is sensitive to the choice of the noise scheduler.

Figure 3 illustrates how the reconstruction loss and the accuracy of the predicted tokens depend on the timestep t for our noise scheduler. In other words, we evaluate how closely the prediction $\hat{\mathbf{x}}_0 = f_\theta(\mathbf{p}_t, \mathbf{0}, t)$ matches the original $\mathbf{x}_0$. Accuracy is calculated only for non-padding tokens.

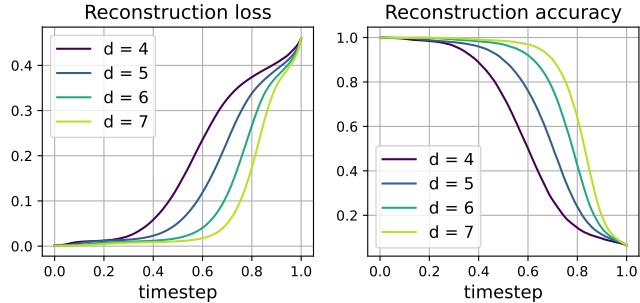

*Figure 3.* Reconstruction loss (left) and reconstruction accuracy (right) w.r.t. timestep for SMOOTHIE, trained with *arctan* noise scheduler with $d = 5$.

## I. Training Dynamics

In this section, we examine the differences in the training dynamics of SMOOTHIE, embedding and simplex diffusions on the ROCStories dataset. Figure 4 illustrates how the Mauve score changes with respect to training time. For embedding diffusion, we perform $1.75\times$ more generation steps (350 vs 200) than for the other diffusion types to match the generation time. The results suggest that, although SMOOTHIE trains and generates text more slowly than the embedding diffusion, it significantly outperforms other methods throughout the training process.

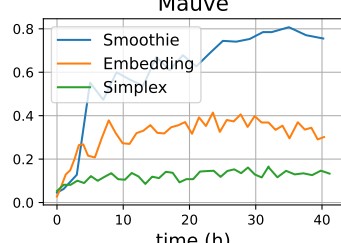

## J. Implementation Details

The hyperparemeters for training and inference of the models across all datasets are presented in Table 12. We trained our models using two 80 GB NVIDIA A100 GPUs for 15 hours on average. For all the tasks, we save checkpoints every 25,000 steps. We select the best checkpoint by the quality on the development set. During generation we do not apply the clamping trick (Li et al., 2022), since it does not improve quality in our experiments. We do not use the classifier-free guidance (Ho & Salimans, 2021) for the same reason.

*Figure 4.* Training dynamics of SMOOTHIE, embedding and simplex diffusions on the ROCStories dataset.

*Table 12.* Complete hyperparameter configurations for all datasets.

| Hyperparameter | ROCStories | XSum | Quasar-T | QQP | ParaDetox |
|---|---|---|---|---|---|
| Tokenizer | | | `bert-base-cased` | | |
| Transformer Layers | | | 12 | | |
| Transformer Dim | | | 768 | | |
| Self-Attention Heads | | | 12 | | |
| Optimizer | | | AdamW | | |
| Learning Rate | | | $2 \cdot 10^{-4}$ | | |
| $\beta_1, \beta_2$ | | | 0.9, 0.98 | | |
| Warmup steps | | | 5000 | | |
| LR scheduler | | | Constant | | |
| Weight decay | | | 0.01 | | |
| Gradient clipping | | | 1 | | |
| EMA decay | | | 0.9999 | | |
| Batch size | 256 | 256 | 512 | 256 | 256 |
| Training steps | 1M | 225k | 150k | 50k | 150k |
| Max input length | — | 512 | 100 | 50 | 40 |
| Max target length | 80 | 64 | 50 | 50 | 40 |
| Generation steps | 200 | 200 | 100 | 100 | 100 |
| $d$ | 5 | 5 | 7 | 5 | 7 |
| $\delta, \sigma_{\min}, \sigma_{\max}$ | | | 1, 1.5, 200 | | |
| $\tilde{\delta}$ | 1.1 | 0.1 | 0.1 | 0.1 | 0.1 |

# K. Dataset Statistics

**ROCStories** The ROCStories dataset (Mostafazadeh et al., 2016) contains 98,161 five-sentence commonsense fictional stories that capture causal and temporal relations between everyday events. It is a widely used small-scale benchmark for unconditional text generation. The dataset is split into 93,161 training instances, 4,000 validation instances, and 1,000 test instances. Url: `https://cs.rochester.edu/nlp/rocstories/`

**XSum** The XSum dataset (Narayan et al., 2018) is used for extreme summarization of BBC news articles. Each article covers a diverse range of topics (e.g., sports, politics) and is paired with a single-sentence summary. The dataset is divided into 204,045 training, 11,332 validation, and 11,334 test instances. Url: `https://huggingface.co/datasets/EdinburghNLP/xsum`

**Quasar-T** Quasar-T (Dhingra et al., 2017) is a large-scale dataset for the question generation task. It requires models to comprehend natural language queries and extract answers from a large corpus. The dataset consists of open-domain trivia questions and their corresponding answers, collected from various internet sources. We use the version preprocessed by Gong et al. (2023a), which includes 116,953 training instances, 2,048 validation instances, and 10,000 test instances. Url: `https://github.com/Shark-NLP/DiffuSeq/tree/main`

**QQP** The Quora Question Pairs (QQP) dataset (Chen et al., 2017) consists of over 400,000 question pairs from the Quora platform, each annotated with a binary label indicating whether the two questions are paraphrases. For the paraphrase generation task, we use the subset containing 149,263 positively labeled pairs, split into 119,410 training instances, 14,926 validation instances, and 14,927 test instances. Url: `https://huggingface.co/datasets/nyu-mll/glue/viewer/qqp`

**ParaDetox** We use ParaDetox dataset (Logacheva et al., 2022) for small-scale conditional generation. It comprises 19,766 pairs of toxic and neutral comments and is intended for the text detoxification task. Url: `https://huggingface.co/datasets/s-nlp/paradetox`

*Table 13.* An example of text semantic degradation obtained with a forward process (Eq. 3).

| | |
|---|---|
| $t = 0$ | My friend Jim **seemed happily married**. He and his wife had **three sons** and seemedate. They left **our area** and moved to **Illinois**. A year later I found out they had gotten divorced. I was shocked and surprised. |
| $t = 0.6$ | My friend Jim **married**. He and his wife had **a son** and feltate. They left **their city** and moved to **California**. A year later I found out she had gotten divorced. He was shocked and happy. |
| $t = 0.7$ | My friend Jim **was getting married**. He and his wife used his daughter in my. They stopped the the and wanted to. The year later he found out he had gotten divorced. He was **excited** and happy. |
| $t = 0.8$ | My friend the the,. He and his wife had a brother of the. She asked the the and wanted. She.. was the the. the. He were happy to him. |
| $t = 0.9$ | John was a the the in the the.. She was the the, the the. He was the the the the. He. the the the.. the. she the the the the. |

## L. Semantic degradation examples

In Table 13, we provide an example showing how decoded text degrades as the noise level $t$ increases. Note that since the diffusion operates in latent space, these decoded texts are approximate. We obtained them by decoding the model predictions for noisy latents sampled with a forward process at each timestep. Nevertheless, the semantic degradation pattern is clearly visible. At low noise levels, the model first substitutes semantically similar content (numbers, synonyms, related proper nouns), then structural coherence degrades, and finally text collapses into high-frequency tokens.

