# OpenReview forum: "Smoothie: Smoothing Diffusion on Token Embeddings for Text Generation"
_ICML.cc/2026/Conference — ICML 2026 regular_

### Official Review · Reviewer_chLg · 2026-03-09

**Soundness:** 3
**Presentation:** 3
**Significance:** 2
**Originality:** 3
**Overall Recommendation:** 4
**Confidence:** 4

**Summary:**

The paper proposes the idea of smoothing diffusion, where text generation is done via continuous noising and denoising. In contrast to simplex diffusion (which neglects the relationship between tokens) and embedding diffusion (which doesn't have a discreteness prior), the paper explores a diffusion process, where:
(1) For every $t$, each token's noisy state is a distribution across tokens;
(2) The logits of the distribution starts from pairwise distance at $t=0$, and gradually became more and more uniform.
The denoising model conditions on the embedding of the distribution, and predicts the embedding of the underlying token.
Experiments are conducted on several datasets.

**Compliance With Llm Reviewing Policy:**

Affirmed.

**Final Justification:**

Empirically, the method doesn't demonstrate a better PPL / DiV tradeoff on OpenWebText. I will maintain my score.

**Key Questions For Authors:**

1. The method depends on a pretrained BERT space. How does the performance rely on the embedding space?
2. The experiments are only conducted on small datasets. Wondering what's the performance on more realistic langauge tasks.
3. On Quasar-T dataset, the method seems to underperform simplex diffusion. Wondering what are potential reasons for that?
4. It's unclear how the baselines (simplex / embedding) diffusion are conducted. Does embedding diffusion use the same x-prediction loss, rather than v-prediction or $\epsilon$-prediction? I would imagine x-prediction to work better in such high dimensional space.

**Limitations:**

yes

**Strengths And Weaknesses:**

Strengths:
1. The experiments are sound and valid. The presentation is overall easy to follow.
2. The proposed method differs from simplex diffusion and embedding space diffusion, which provides a new way to do continuous generation.
3. Some improvements are shown, compared to simplex or embedding diffusions.

Weaknesses:
1. The method depends on a pretrained embedding space, so is somewhat non self-contained.
2. The experiments are only conducted on small datasets. It's unclear how the method scales to more realistic langauge datasets.
3. The generation and training seems to be slower, compared to pure embedding space generation.

---

> ### Author Rebuttal · Authors · 2026-03-31
>
> We thank all reviewers for their constructive and insightful feedback. We are encouraged that reviewers found our theoretical foundation "elegant" (PAff) and "robust" (roVG), the method "novel" providing "a new way to do continuous generation" (chLg), and the experiments "sound and valid" (chLg). Below we address the raised concerns.
>
> ## W3: Computational complexity
>
> SMOOTHIE is indeed slower than the embedding diffusion. However, we want to clarify two important points about the computational complexity.
>
> **SMOOTHIE has the same complexity as simplex diffusion.** As stated in Section 5 and Table 2, SMOOTHIE has exactly the same computational complexity as simplex diffusion methods (SSD-LM, TESS), which also require an $\mathcal{O}(B×L×d×V)$ linear head at every step, while surpassing them in quality. In addition to this, the complexity is also the same for the majority of Masked Diffusion models [1, 2].
>
> **Potential optimizations exist.** As we discussed in Appendix A, these include (1) reducing the effective vocabulary size; (2) clustering embeddings for all timesteps and performing smoothie over the cluster centroids. Additionally, we can implement consistency distillation to reduce the number of generation steps.
>
> ## Q1: Dependence on pre-trained embeddings
>
> Reliance on a fixed embedding matrix is a common design choice across text diffusion models [3, 4]. The ablation in Table 8 (Appendix F) shows that even with GPT-2 embeddings, SMOOTHIE achieves MAUVE 64.4, which still outperforms most diffusion baselines in Table 4. This demonstrates reasonable robustness to embedding choice, although some embeddings are better than others.
>
> Regarding end-to-end training: we attempted joint training of embeddings with the diffusion model but observed embedding collapse under pure MSE loss. Adding cross-entropy loss prevented collapse but caused embedding norm explosion. We tried gradient clipping, separate learning rates, and periodic re-normalization, but none reliably stabilized training. This is discussed in Appendix A, and we believe solving this is an important but orthogonal research direction. We will summarize these findings in the main text.
>
> ## Q2: Small datasets
>
> We acknowledge that the main experiments are conducted on relatively small datasets, consistent with the evaluation protocol used by all prior text diffusion works (DiffuSeq, TESS, TEncDM, SSD-LM) to enable direct comparison.
>
> To address this concern, we conducted additional experiments on the **OpenWebText** dataset with sequence length 512, which is a more realistic benchmark. We compare SMOOTHIE to our reimplementation of Embedding diffusion and GIDD [5], a strong masked diffusion model.
>
> | Method | MAUVE ↑ | PPL ↓ | Div ↑ |
> |---|---|---|---|
> | GIDD | 28.6 | 228.3 | 58.8 |
> | Embedding diffusion | 55.7 | 51.9 | 21.2 |
> | **SMOOTHIE (ours)** | **59.9** | 62.2 | 23.4 |
>
> Same as in paper, here we rely primarly on the Mauve metric, because perplexity can be low if the diversity is low and text contains repetitions, while high diversity may be a result of the presence of random words. Mauve, on the other hand, directly measures the proximity of generated and reference text distributions. SMOOTHIE outperforms all other models, confirming that the advantages of our approach hold on larger-scale data with longer sequences. Note that GIDD's high diversity comes at the cost of extremely poor perplexity (228.3). We will include these results in the revised manuscript.
>
> ## Q3: Underperformance on Quasar-T vs simplex diffusion.
>
> The differences between SMOOTHIE and Simplex diffusion on Quasar-T are very small (BERTScore 63.1 vs 63.0, BLEU 19.9 vs 19.3, R-L 36.5 vs 36.9), so it is hard to say which model is better. Quasar-T is a particularly noisy dataset with sometimes incorrect examples. Therefore, it is hard to analyze the reasons of this behaviour.
>
> ## Q4: Prediction type for embedding diffusion baseline.
>
> Yes, same as SMOOTHIE, embedding diffusion is trained to predict a clean sample $x_0$. All text diffusion papers argee that it works better than predicting noise $\varepsilon$.
>
> [1] J. Austin et al. Structured Denoising Diffusion Models in Discrete State-Spaces // NeurIPS 2021
> [2] S. Sahoo et al. Simple and Effective Masked Diffusion Language Models // NeurIPS 2024
> [3] V. Meshchaninov et al. COSMOS: Compressed and Smooth Latent Space for Text Diffusion Modeling // NeurIPS 2025
> [4] A. Shabalin et al. TEncDM: Understanding the Properties of the Diffusion Model in the Space of Language Model Encodings // AAAI 2025
> [5] D. Rütte et al. Generalized Interpolating Discrete Diffusion // ICML 2025

---

> > ### Author Rebuttal · Reviewer_chLg · 2026-04-04
> >
> > I thank the author for the rebuttal. It partially addressed my concerns.
> >
> > On the additional experiments, it's unclear that the proposed method outperform embedding diffusion baseline, since gen PPL is worse (so it might be achieving a different PPL vs diversity tradeoff). Also, the smoothie method is slower than embedding diffusion, as shown in table 2 of the paper. I will keep my score.

---

> > > ### Author Response · Authors · 2026-04-05
> > >
> > > We thank Reviewer chLg for the follow-up. We would like to address the remaining concerns.
> > >
> > > ### On the PPL vs. diversity trade-off
> > >
> > > As discussed in Section 5.1 of our paper, the hyperparameter $\tilde{\delta}$ allows explicit control over the trade-off between perplexity and diversity. To demonstrate that SMOOTHIE outperforms embedding diffusion at a comparable perplexity operating point, we ran an additional experiment with a lower $\tilde{\delta}$ (0.98 instead of 1.02):
> > >
> > > |Method|MAUVE ↑|PPL ↓|Div ↑|
> > > | --- | --- | --- | --- |
> > > |GIDD|28.6|228.3|58.8|
> > > |Embedding diffusion|55.7|51.9|21.2|
> > > |SMOOTHIE ($\tilde{\delta} = 1.02$)|59.9|62.2|23.4|
> > > |SMOOTHIE ($\tilde{\delta} = 0.98$)|58.9|48.2|19.8|
> > >
> > > Here, with lower perplexity, SMOOTHIE still surpasses an Embedding diffusion in terms of MAUVE score. Therefore, SMOOTHIE's advantage is not simply a different trade-off point. It genuinely produces text that is closer to the reference distribution.
> > >
> > > ### On generation speed
> > > As noted in Table 2 of our paper, embedding diffusion is 1.75× faster per step than SMOOTHIE. To ensure a fair comparison, we give embedding diffusion 1.75× more generation steps in all experiments, so that the total wall-clock generation time is matched. The results above are obtained under this matched-time protocol, meaning SMOOTHIE's higher MAUVE score is achieved at the same computational budget.

---

### Official Review · Reviewer_roVG · 2026-03-09

**Soundness:** 3
**Presentation:** 3
**Significance:** 2
**Originality:** 3
**Overall Recommendation:** 4
**Confidence:** 3

**Summary:**

This paper proposes SMOOTHIE, a diffusion framework for text generation that operates in a semantically informed, discrete-compatible latent space. In this framework, each token is represented by its vector of negative squared distances to all vocabulary embeddings; the forward process progressively smooths these distances using a Gaussian kernel bandwidth schedule. Decoding remains natural via a softmax over the distance logits. The authors provide a training objective that predicts clean embeddings and introduce a simple reverse solver that ignores the previous latent state. Across one unconditional and four sequence-to-sequence (seq2seq) benchmarks, SMOOTHIE consistently outperforms prior diffusion-based text generators; under matched architectures, it also improves over models using both embedding-based and simplex-based diffusion spaces.

**Compliance With Llm Reviewing Policy:**

Affirmed.

**Final Justification:**

The author resolved my doubts, and I'm willing to give a higher rating.

**Key Questions For Authors:**

1.Given that performance is highly sensitive to embedding quality, what specific regularization techniques were attempted to stabilize this process?

2. Is there any preliminary experimental evidence indicating how SMOOTHIE's advantages over autoregressive models might change as the parameter scale increases?

3. Could the authors provide a concrete qualitative example in the main text to intuitively demonstrate that the probability mass indeed diffuses toward semantically related tokens first, before spreading to unrelated ones?

4. The paper transparently notes that embedding diffusion is 1.75x faster than SMOOTHIE during generation. For potential real-world applications where inference latency is critical, do the authors have any brief recommendations or future directions for engineering optimizations to close this speed gap?

5. The authors provide a very honest and insightful discussion of the model's limitations in Appendix A. Given the importance of these constraints for future researchers building upon this work, would it be possible to summarize these key limitations in the Discussion or Conclusion section of the main text?

If the author can solve my doubts, I can raise the score.

**Limitations:**

Yes

**Strengths And Weaknesses:**

Strengths：

1. The paper is structured well. The paper introduces a novel smoothing diffusion framework that bridges the gap between continuous Gaussian diffusion and discrete categorical simplex methods.

2. The authors provide a robust theoretical foundation, proving that the distance-based latent space generalizes previous simplex-based models. Specifically, they demonstrate that simplex diffusion is mathematically equivalent to a SMOOTHIE model using a trivial metric.

3. Optimizing MSE on clean embeddings is provably equivalent to optimizing MSE on distance logits,this simplifies learning while preserving optimality with respect to the forward model.

4. The paper consistently outperforms state-of-the-art diffusion baselines across various tasks, including unconditional generation, summarization, and detoxification.

5. It consistently achieves stable improvements over diffusion baseline models across multiple seq2seq tasks.

Weaknesses：

1. The paper exhibits high computational complexity because it requires calculating pairwise distances between predicted embeddings and all embeddings in the vocabulary at every generation step. Compared to basic embedding-based diffusion models, SMOOTHIE’s generation speed is significantly slower.

2. The framework's quality relies heavily on a fixed BERT embedding matrix. Ablation studies indicate that performance is sensitive to embedding quality.

3. While the authors provide an honest evaluation of the bottlenecks (such as complexity and training stability), these critical discussions are relegated to Appendix A rather than being integrated or summarized within the main discussion or conclusion sections.

4. The generation process is constrained by a predefined maximum length, failing to truly sample variable-length sequences as autoregressive models do.

---

> ### Author Rebuttal · Authors · 2026-03-31
>
> We thank all reviewers for their constructive and insightful feedback. We are encouraged that reviewers found our theoretical foundation "elegant" (PAff) and "robust" (roVG), the method "novel" providing "a new way to do continuous generation" (chLg), and the experiments "sound and valid" (chLg). Below we address the raised concerns.
>
> ## W1, Q4: Computational complexity
>
> We want to clarify two important points about the computational complexity.
>
> **SMOOTHIE has the same complexity as simplex diffusion.** As stated in Section 5 and Table 2, SMOOTHIE has exactly the same computational complexity as simplex diffusion methods (SSD-LM, TESS), which also require an $\mathcal{O}(B×L×d×V)$ linear head at every step, while surpassing them in quality. In addition to this, the complexity is also the same for the majority of Masked Diffusion models [1, 2].
>
> **Potential optimizations exist.** As we discussed in Appendix A, these include (1) reducing the effective vocabulary size; (2) clustering embeddings for all timesteps and performing smoothie over the cluster centroids. Additionally, we can implement consistency distillation to reduce the number of generation steps.
>
> ## W4: Predefined maximum length
>
> We would like to argue that SMOOTHIE can indeed sample variable-length sequences. Our method is trained to sample texts from the data distribution. This means that the length distribution of the generated texts is dynamic and matches that of the source data. In all text diffusion models, it is implemented by generating PAD tokens and discarding them once the text is complete. It is important to note, that AR models are also limited in the text length by the context size used during the training in the same way as diffusion models.
>
> ## Q1: Dependence on pre-trained embeddings
>
> Reliance on a fixed embedding matrix is a common design choice across text diffusion models [3, 4]. The ablation in Table 8 (Appendix F) shows that even with GPT-2 embeddings, SMOOTHIE achieves MAUVE 64.4, which still outperforms most diffusion baselines in Table 4. This demonstrates reasonable robustness to embedding choice, although some embeddings are better than others.
>
> Regarding end-to-end training: we attempted joint training of embeddings with the diffusion model but observed embedding collapse under pure MSE loss. Adding cross-entropy loss prevented collapse but caused embedding norm explosion. We tried gradient clipping, separate learning rates, and periodic re-normalization, but none reliably stabilized training. This is discussed in Appendix A, and we believe solving this is an important but orthogonal research direction. We will summarize these findings in the main text.
>
> ## Q2: Advantages over autoregressive models with scaling
>
> The mentioned advantages of diffusion models over AR models are preserved at every parameter scale because they are built in the model design. Namely, diffusion models are not restricted by the left-to-right one-token-at-a-time generation pattern. This unlocks capabilities such as parallel decoding, iterative refinement, and constrained generation (e.g., infilling).
>
> ## Q3: Qualitative example of semantic probability diffusion.
>
> We provide an example below showing how decoded text degrades as the noise level t increases. Note that since the diffusion operates in latent space, these decoded texts are approximate. We obtained them by decoding the model predictions for noisy latents at each timestep. Nevertheless, the semantic degradation pattern is clearly visible:
>
> **Example:**
> - t=0: *My friend Jim seemed happily married. He and his wife had three sons.... They left our area and moved to Illinois.*
> - t=0.6: *...had a son and feltate. They left their city and moved to California.* (Semantic substitutions: "three sons" → "a son", "our area" → "their city", "Illinois" → "California")
> - t=0.7: *He and his wife used his daughter in my. They stopped the the and wanted to.* (Semantic structure breaks)
> - t=0.8–0.9: *John was a the the in the the.. She was the the..* (Collapse to repetitive words)
>
> The pattern is consistent: at low noise levels, the model first substitutes semantically similar content (numbers, synonyms, related proper nouns), then structural coherence degrades, and finally text collapses into high-frequency tokens. We will include these examples in the revised paper.
>
> ## Q5: Moving limitations to main text.
>
> We agree that key limitations should be accessible in the main text. We will add a limitations paragraph to the Conclusion section summarizing the discussion currently in Appendix A.
>
> [1] J. Austin et al. Structured Denoising Diffusion Models in Discrete State-Spaces
> [2] S. Sahoo et al. Simple and Effective Masked Diffusion Language Models
> [3] V. Meshchaninov et al. COSMOS: Compressed and Smooth Latent Space for Text Diffusion Modeling
> [4] A. Shabalin et al. TEncDM: Understanding the Properties of the Diffusion Model in the Space of Language Model Encodings

---

> > ### Author Rebuttal · Reviewer_roVG · 2026-04-07
> >
> > The author resolved my doubts, and I'm willing to give a higher rating.

---

### Official Review · Reviewer_PAff · 2026-03-10

**Soundness:** 2
**Presentation:** 3
**Significance:** 2
**Originality:** 2
**Overall Recommendation:** 4
**Confidence:** 2

**Summary:**

The paper introduces SMOOTHIE, a novel diffusion framework for text generation that seeks to reconcile the discrete nature of text with the continuous, semantic structure of token embedding. The authors evaluate the model on unconditional and sequence-to-sequence text generation tasks, demonstrating performance improvements over existing text diffusion baselines using a custom sampling solver.

**Compliance With Llm Reviewing Policy:**

Affirmed.

**Final Justification:**

Honestly speaking, I am not familiar with this field, but the authors have resolved my issues, I will raise my score.

**Key Questions For Authors:**

1. By removing the previous state from the update step (Eq. 9), you are effectively performing direct x_0 prediction and re-noising, stripping the momentum inherent to the reverse diffusion ODE/SDE. Can you provide a more rigorous mathematical justification for why this is valid within the diffusion framework, rather than simply stating it empirically prevents "drifting"?

2. The distance matrix computation requires calculating distances to every token in the vocabulary at every step. Have you profiled the generation time and peak VRAM usage for modern vocabulary sizes (e.g., LLaMA's 128k tokenizer) compared to standard embedding diffusion? How do you propose to scale this operation mathematically?

3. Given that SMOOTHIE requires 100-200 steps and barely matches or slightly underperforms BART/GPT-2, what is the practical incentive to deploy this model? Are there specific controllable generation tasks where SMOOTHIE natively succeeds but AR models fail?

**Limitations:**

No.

Constructive suggestions for improvement:

1. The authors need to explicitly discuss the scaling limitations of their approach in the main text. Current evaluations are limited to ~100M parameter models, whereas the field has moved to billion-parameter scales.

**Strengths And Weaknesses:**

Strengths:

1. The research's notable contribution concerns the fusion of categorical simplex diffusion with continuous embedding spaces via distance matrices.

2. The mathematical formulation bridging continuous and discrete diffusion is elegant. Specifically, the proof in Appendix B demonstrating that simplex-based diffusion is a special case of SMOOTHIE's distance-based latent space (under a trivial metric) provides a strong theoretical foundation for the approach.

Weaknesses:

1. Despite requiring 100-200 generation steps (which is extremely slow due to the O(V) distance calculation), SMOOTHIE merely matches or slightly trails standard autoregressive models (like BART or GPT-2, Table 4) that generate text in a fraction of the time. The paper does not sufficiently demonstrate a unique advantage (e.g., zero-shot control, non-left-to-right constrained generation) that justifies the extreme inference cost over standard AR baselines.

2. The practical significance of the method is currently limited by severe computational complexity. During generation, computing pairwise distances to all vocabulary embeddings at every timestep scales as O(B×L×d×V). For modern LLM vocabularies (e.g., 50k - 128k tokens), this O(V) dependency per step is a massive bottleneck, fundamentally limiting the method's scalability beyond small vocabularies and models.

3. The proposed reverse sampling solver (Eq. 9) is mathematically concerning. The authors explicitly state that they drop the previous latent state (D_{t−1}) when computing the update. While Theorem 4.1 relates the optimal predictions of g_\theta and f_\theta, discarding the ancestral state fundamentally breaks the Markovian chain structure of the standard reverse diffusion process (DDPM/DDIM). It reduces the method to an iterative denoising/refinement mechanism rather than a principled diffusion sampler. Furthermore, the empirical justification for this deviation (claiming "sparsity" causes drift) is speculative and lacks rigorous theoretical analysis.

---

> ### Author Rebuttal · Authors · 2026-03-31
>
> We thank all reviewers for their constructive and insightful feedback. We are encouraged that reviewers found our theoretical foundation "elegant" (PAff) and "robust" (roVG), the method "novel" providing "a new way to do continuous generation" (chLg), and the experiments "sound and valid" (chLg). Below we address the raised concerns.
>
> ## W1-2: Computational complexity and scalability
>
> We want to clarify two important points about the computational complexity.
>
> **SMOOTHIE has the same complexity as simplex diffusion.** As stated in Section 5 and Table 2, SMOOTHIE has exactly the same computational complexity as simplex diffusion methods (SSD-LM, TESS), which also require an $\mathcal{O}(B×L×d×V)$ linear head at every step, while surpassing them in quality. In addition to this, the complexity is also the same for Duo model [1] and the majority of Masked Diffusion models [2, 3].
>
> **Potential optimizations exist.** These include embedding clustering at each timestep, vocabulary reduction, and distillation techniques to reduce the number of generation steps (as explored in image diffusion). We will add a discussion of scaling considerations and engineering optimizations to the main text.
>
> ## Q1: Justification for dropping the previous state (Eq. 9).
>
> We appreciate your concern on this topic. Our proposed sampler is indeed "incorrect" in the strict sense that it does not reproduce the forward process trajectory. In a variance-preserving scheme, the variance of the i-th coordinate satisfies $Var_i(x_t)=1$, whereas our solver yields $Var_i(x_{t−1})=ᾱ_{t−1}Var_i(f(x\_t, t))+1−ᾱ_{t−1}<1$. However, the reverse process is not required to perfectly match the forward dynamics, it should only match the data distribution at $t=0$. Upon further analysis we identified two concrete reasons why this deviation is beneficial in practice.
>
> **First, for sequence-to-sequence tasks the deviation is small.** In conditional generation, $Var_i(f(x_t, t, w^x))$ is close to 1 for the majority of timesteps, because the conditioning input significantly restricts the space of possible outputs. Therefore, while omitting the previous latent state does change the trajectory dynamic, this change is not drastic.
>
> **Second, the reduced variance acts as a conservative correction in sparse latent spaces.** The latent space built on shallow token embeddings is highly sparse, because at low noise levels, the probability mass is concentrated near discrete vocabulary token representations. In such a regime, even small drift in an incorrect direction can push the latent out of distribution, causing compounding errors in subsequent steps. Since $Var_i(x_{t−1})<1$, our solver effectively "commits" less to any single prediction compared to DDPM. When the diffusion model makes an inaccurate prediction, DDPM may produce out-of-distribution latents that spoil future steps, whereas our solver stays closer to the center of the space where the data density is higher, providing a form of implicit regularization.
>
> Additionally, the DDIM comparison (see Reviewer jpGm Q1) provides empirical evidence that DDIM also underperforms our approach across all tasks. We will expand this analysis in the revised manuscript.
>
> ## Q2: Profiling for large vocabularies
>
> We have not yet profiled specifically with 128k vocabularies, but note that the distance computation is a matrix multiplication of shape (B×L, d) × (d, V), identical in structure to the final linear projection in simplex diffusion and in the language model head of any standard LLM.
>
> ## Q3: Practical incentive over AR models.
>
> At the current state of research, AR models remain superior in both speed and quality for standard text generation tasks. However, the primary contribution of this paper is advancing diffusion-based text generation, not replacing AR models. SMOOTHIE introduces a principled approach to diffusion model design that outperforms all existing diffusion methods. We see this as evidence that the gap between diffusion and AR models is closing, and that designing a noising process better suited to text, which we propose, seems like a key ingredient. Regarding practical applicability, all text diffusion models employ bidirectional attention, which provides an advantage for tasks such as text infilling, as demonstrated by [4].
>
> ## Suggestion: Discuss scaling limitations in main text.
>
> We argee that this is an important detail. We will add a discussion of scaling considerations to the main text and include the OpenWebText results (see Reviewer chLg Q2) as evidence that our method works on larger-scale data.
>
> [1] S. Sahoo et al. The Diffusion Duality // ICLR 2025 Workshop
> [2] J. Austin et al. Structured Denoising Diffusion Models in Discrete State-Spaces // NeurIPS 2021
> [3] S. Sahoo et al. Simple and Effective Masked Diffusion Language Models // NeurIPS 2024
> [4] X. Li et al. Diffusion-LM Improves Controllable Text Generation // NeurIPS 2022

---

> > ### Author Rebuttal · Reviewer_PAff · 2026-04-01
> >
> > Honestly speaking, I am not familiar with this field, but the authors have resolved my issues, I will raise my score.

---

### Official Review · Reviewer_jpGm · 2026-03-12

**Soundness:** 2
**Presentation:** 3
**Significance:** 3
**Originality:** 3
**Overall Recommendation:** 4
**Confidence:** 3

**Summary:**

The paper introduces SMOOTHIE, a text diffusion framework that applies a smoothing mechanism to token embeddings based on semantic similarities. By representing tokens through their distance vectors relative to the entire vocabulary, the authors bridge the gap between discrete categorical diffusion and continuous semantic space. The work also proposes a non-Markovian sampling strategy that demonstrates empirical gains over the standard DDPM solver.

**Compliance With Llm Reviewing Policy:**

Affirmed.

**Key Questions For Authors:**

1. The sampling method proposed in the paper is a non-Markovian method, somewhat similar to DDIM. However, the paper only compares this method with the DDPM solver as one of its contributions. Could you provide more discussion or experimental verification regarding the comparison with DDIM?
2. Some experimental metrics did not demonstrate the superior performance of the proposed method. For example, on the ROCStories dataset, MDLM significantly outperformed this method in the Div index. Do these relatively poor performance metrics reveal limitations of the method, and could you explain them?

**Limitations:**

yes

**Strengths And Weaknesses:**

Strengths:
1. The proposed method defines the diffusion process on the Euclidean distance vector of the token embedding. This design combines the advantages of continuous diffusion models and discrete/simplex diffusion models, overcoming the limitations of previous methods in semantic degradation and token decoding.
2. An improved sampler is proposed that ignores the previous state when updating the hidden state. Experiments show that this is more effective than the traditional DDPM sampler when dealing with sparse distributions of text embeddings.

Weaknesses:
1. This method requires calculating the distance matrix between each token and the entire vocabulary, resulting in slow processing speed and high computational overhead. While the authors acknowledge this limitation and intuitively dismiss approximations such as top-k truncation, the manuscript lacks empirical evidence to quantify the performance-efficiency trade-off or to identify the boundary of performance degradation. Therefore, the significant efficiency hurdle currently limits the method's practicality, and the viability of potential optimizations is left as an open question.
2. The theoretical foundation of the proposed improved sampler is somewhat underdeveloped. As a non-Markovian sampling approach, it shares significant conceptual similarities with DDIM. However, the experimental section only evaluates its performance against the standard DDPM, which is a Markovian baseline. Without a direct comparison or a rigorous theoretical analysis highlighting its distinct advantages over DDIM, it is difficult to assess the actual contribution of this sampler.
3. There are some data gaps in the experimental comparisons. Furthermore, the selected AR models are relatively traditional (such as FLAN-T5 and GPT-2), lacking more advanced AR models. These factors affect the comprehensiveness and fairness of the method evaluation.
4. The paper claims that the proposed method can also be applied to other categories where there are semantic relationships between categories (e.g. graphics, protein sequences), but the authors do not provide sufficient analysis and evidence to support this claim.

---

> ### Author Rebuttal · Authors · 2026-03-31
>
> We thank all reviewers for their constructive and insightful feedback. We are encouraged that reviewers found our theoretical foundation "elegant" (PAff) and "robust" (roVG), the method "novel" providing "a new way to do continuous generation" (chLg), and the experiments "sound and valid" (chLg). Below we address the raised concerns.
>
> ## W1. Performance-efficiency trade-off for top-k truncation
>
> We would like to provide empirical evidence that top-k truncation is not a viable optimization. At almost all timesteps, a large proportion of embeddings carry significant weight after softmax, while top-k truncation only provides a speed boost when a very small number of closest vectors need to be extracted.
>
> We calculated the minimum fraction of vocabulary embeddings whose combined weight exceeds a threshold p, for various timesteps t:
>
> |p|t=0.01|t=0.05|t=0.1|t=0.2|t=0.5|t=0.75|t=0.99|
> |---|---|---|---|---|---|---|---|
> |0.90|0.001|0.003|0.29|0.52|0.60|0.60|0.60|
> |0.95|0.001|0.008|0.43|0.66|0.73|0.73|0.73|
> |0.99|0.001|0.066|0.70|0.86|0.90|0.90|0.90|
>
> The results show that to achieve a combined weight of p=0.9, more than half of all embeddings must be retained for almost all timesteps (t ≥ 0.2). This confirms that top-k truncation cannot meaningfully accelerate inference without substantially degrading accuracy. We will include this analysis in the revised manuscript.
>
> Additionally, we want to emphasise that SMOOTHIE has exactly the same computational complexity as simplex diffusion methods (SSD-LM, TESS), which also require an $\mathcal{O}(B×L×d×V)$ linear head at every step, as stated in Section 5 and Table 2. Moreover, the complexity is also the same for Duo model [1] and the majority of Masked Diffusion models [2, 3].
>
> ## W3. Comparison comprehensiveness
>
> We agree that the gaps in the results tables may be confusing. We aimed to include as many models as possible and took reported results from the corresponding papers, which is why some metrics are missing. For AR baselines, we chose the most popular models with a similar number of parameters (~100M) to provide a fair understanding of how our non-autoregressive approach compares. We see that while being close, SMOOTHIE still mostly loses to AR methods. However, our primary focus is on advancing diffusion-based generation, and SMOOTHIE surpasses all diffusion baselines.
>
> ## W4. Application to other domains
>
> We agree that the claim about applicability to other domains (e.g., graphs, protein sequences) is not experimentally validated in this work and is stated as a future direction. The theoretical basis for this claim is straightforward: SMOOTHIE's framework (Eq. 13 in Appendix B) is defined for any categorical domain equipped with a distance metric $\rho(w, i)$ between categories. For example, we can use BLOSUM for protein sequences of  domain-specific embeddings for graphs with categorical node/edge types. We acknowledge that the choice of an appropriate metric for each domain requires careful investigation, and we have stated this explicitly in Section 6. We will soften the claim in the revised manuscript to more clearly frame it as a motivated direction rather than a validated capability.
>
> ## Q1. Comparison with DDIM
>
> Thank you for this excellent question. Our solver and DDIM are conceptually related as both are non-Markovian, but they differ in an important way. DDIM interpolates between the current latent and the predicted clean sample, retaining dependence on the previous state. Our solver (Eq. 9) completely discards the previous latent and re-noises from the predicted clean sample.
>
> We ran experiments with the DDIM solver adapted to our framework:
>
> ||XSum (BS / R-L)|Quasar-T (BS / R-L)|QQP (BS / R-L)|ParaDetox (J-Score)|
> |---|---|---|---|---|
> |DDPM|68.5 / 25.6 |62.8 / 35.5|83.7 / 60.7|51.5|
> |DDIM| 67.4 / 24.6 | 60.3 / 31.8 | 82.0 / 57.9 | 50.2 |
> |Ours|**68.8 / 26.0**|**63.1 / 36.5**|**83.9 / 60.9**|**51.7**|
>
> Our solver consistently outperforms both DDPM and DDIM across all tasks. We provide a more detailed analysis of why this occurs in our response to Reviewer PAff Q1. We will include this comparison in the revised paper.
>
> ## Q2. MDLM outperforming SMOOTHIE on Diversity (ROCStories)
>
> High diversity alone is not desirable — random text would maximize diversity. The MAUVE score, which directly measures similarity between generated and reference distributions, is a more meaningful metric. MDLM achieves higher diversity (35.1 vs 25.1) but at the cost of substantially worse perplexity (58.1 vs 24.9) and MAUVE score (63.9 vs 80.7). This indicates that MDLM's high diversity comes from generating less coherent text, not from meaningful lexical variety.
>
> [1] S. Sahoo et al. The Diffusion Duality // ICLR 2025 Workshop
> [2] J. Austin et al. Structured Denoising Diffusion Models in Discrete State-Spaces // NeurIPS 2021
> [3] S. Sahoo et al. Simple and Effective Masked Diffusion Language Models // NeurIPS 2024

---

> > ### Author Rebuttal · Reviewer_jpGm · 2026-04-03
> >
> > The rebuttal provides additional analysis and clarifications, but it does not address all of my concerns, so I maintain my score.

---

### Decision · Program_Chairs · 2026-04-30

**Decision:**

Accept (regular)

**Comment:**

This paper proposes Smoothie, a text diffusion framework that performs diffusion over token embeddings by representing each token via distances to the vocabulary.

During rebuttal, the authors addressed several concerns by providing additional experiments, including comparisons with DDIM and analysis of performance trade-offs, and by clarifying limitations and applicability. Following these clarifications, multiple reviewers indicated that their concerns were resolved or partially resolved and raised or maintained their scores at weak accept.

Remaining concerns include computational complexity and unclear performance improvements under some settings (e.g., trade-offs between perplexity and diversity). However, these concerns do not outweigh the merits of the paper and can be addressed in future work.

Overall, based on the reviewers’ assessments and updates after rebuttal, I recommend acceptance.